# Exposure of *Mycobacterium avium* subsp. *homonissuis* to Metal Concentrations of the Phagosome Environment Enhances the Selection of Persistent Subpopulation to Antibiotic Treatment

**DOI:** 10.3390/antibiotics9120927

**Published:** 2020-12-19

**Authors:** Lia Danelishvili, Elyssa Armstrong, Emily Miyasako, Brendan Jeffrey, Luiz E. Bermudez

**Affiliations:** 1Department of Biomedical Sciences, Carlson College of Veterinary Medicine, Oregon State University, Corvallis, OR 97331, USA; armstrel@oregonstate.edu (E.A.); miyasake@oregonstate.edu (E.M.); 2Bioinformatics and Computational Biosciences Branch, Rocky Mountain Laboratories, National Institute of Allergy and Infectious Diseases, National Institutes of Health, Hamilton, MT 59840, USA; brendan.jeffrey@nih.gov; 3Department of Microbiology, College of Sciences, Oregon State University, Corvallis, OR 97331, USA

**Keywords:** phagosome environment, macrophages, antibiotic susceptibility, persistence, *Mycobacterium avium* subsp. *hominissuis*, host immune response

## Abstract

*Mycobacterium avium* subspecies *hominissuis* (MAH) is an opportunistic intracellular pathogen causing infections in individuals with chronic lung conditions and patients with immune-deficient disorders. The treatment of MAH infections is prolonged and outcomes many times are suboptimal. The reason for the extended treatment is complex and reflects the inability of current antimicrobials to clear diverse phenotypes of MAH quickly, particularly, the subpopulation of susceptible but drug-tolerant bacilli where the persistent fitness to anti-MAH drugs is stimulated and enhanced by the host environmental stresses. In order to enhance the pathogen killing, we need to understand the fundamentals of persistence mechanism and conditions that can initiate the drug-tolerance phenotype in mycobacteria. MAH can influence the intracellular environment through manipulation of the metal concentrations in the phagosome of infected macrophages. While metals play important role and are crucial for many cellular functions, little is known how vacuole elements influence persistence state of MAH during intracellular growth. In this study, we utilized the in vitro model mimicking the metal concentrations and pH of MAH phagosome at 1 h and 24 h post-infection to distinguish if metals encountered in phagosome could act as a trigger factor for persistence phenotype. Antibiotic treatment of metal mix exposed MAH demonstrates that metals of the phagosome environment can enhance the persistence state, and greater number of tolerant bacteria is recovered from the 24 h metal mix when compared to the viable pathogen number in the 1 h metal mix and 7H9 growth control. In addition, bacterial phenotype induced by the 24 h metal mix increases MAH tolerance to macrophage killing in TNF-α and IFN-γ activated cells, confirming presence of persistent MAH in the 24 h metal mix condition. This work shows that the phagosome environment can promote persistence population in MAH, and that the population differs dependent on a concentration of metals.

## 1. Introduction

The non-tuberculous *Mycobacterium avium* subsp. *hominissuis* (MAH) is a human pathogen that poses a high risk of infection in immunocompromised individuals. While MAH infection leads to bacteremia in AIDS and other immune suppressed patients, people with underlying lung conditions such as emphysema, bronchiectasis or cystic fibrosis develop chronic pulmonary infections [1,2]. MAH can colonize and invade mucosal epithelial cells of respiratory tract, but preferably resides inside macrophages. In the phagocytic cells, bacteria establish the survival niche within cytoplasmic vacuoles originated from the plasma membrane [3,4]. The vacuoles do not mature and, therefore, do not fuse with lysosomes [4,5]. In order for antibiotics to reach the bacterial target and exert a killing effect, they must cross several cellular barriers including the semipermeable plasma membrane, the phagosome vacuole layer and the bacterial cell wall. Mycobacteria are characterized with a thick, highly impermeable cell wall that consists of insoluble components such as peptidoglycan, arabinogalactan and mycolic acid. As a result, the concentration of the antibiotics reaching inside the bacterial cell is significantly diminished compared with the serum concentration [6]. Within an intracellular environment of the host and in presence of continuous exposure to less than an inhibitory concentration of antibiotics, bacteria undergo phenotypic remodeling that is characterized with a low metabolic and low growth rates, leading to a non-replicative state. The switch from fast to slow growth is a phenotypic trait independent of genetics and, thus, persistent bacteria that are “resistant” to antibiotics during the course of treatment remain genetically susceptible to drugs. Furthermore, the host environmental stresses and conditions are additional factors contributing to bacterial altered metabolic state, thus, shaping the persistence fitness to anti-mycobacterial drugs. These processes drive a selection of the tolerant subpopulation of mycobacteria that can resist the bactericidal concentration of antimicrobials [7,8]. To successfully kill an intracellular organism that presumably is in a very different metabolic and phenotypic state represents a chief challenge, as the majority of the available antimicrobials targets actively growing bacteria.

Persistence phenomenon has long been studied in *Mycobacterium tuberculosis* [9]. There is a solid experimental evidence in vivo as well as in clinics on *M. tuberculosis* ability to persist in tissues in non-replicative state ranging from months to decades, yet capable returning to growth state later and activating disease. It is a fact that, after infection, 5–10% of individuals develop tuberculosis disease and, after the combinational treatment completion, the post-primary tuberculosis can arise years later from the reactivation of persistent infection. In addition, in asymptomatic individuals, the viable *M. tuberculosis* can be found outside of lesions within the lung tissue or lymph nodes and can also result in post-primary disease [10]. The exact mechanism or trigger behind the epigenetic switch into a persistent state remains unclear, although, it is suggested that macrophage physiology, including intracellular stresses and the environmental condition such as hypoxia, increase chances for an unset of a persistence phenotype. The hypoxia is one of the components of the host inflammatory response, and low oxygen condition has been suggested to stimulate *M. tuberculosis* persistence in mice via activation of the isocitrate lyase enzyme of the fatty acid metabolism [11]. The toxin-antitoxin loci have been reported to be activated under stresses encountered in vivo, including hypoxia and phagocytosis by macrophages, and contributing to mycobacterial persistence [10]. *M. tuberculosis* has 88 toxin-antitoxin genes, many of them associated with persistence [12]. The *hspR* deletion, an important regulatory gene for heat-shock proteins, results in a persistence defect and the virulence regulator PhoP responds to hypoxic stresses by metabolic switch of nitrogen metabolism for long-term survival [13]. In addition, the oxidative stress and reactive oxygen species (ROS) can activate bacterial efflux system for recycling of damaged proteins and, thus, enhancing mycobacterial tolerance to antibiotics by increased drug efflux from bacterial cells [14,15]. 

The persistence phenomenon has been less investigated for MAH [16]. The experimental evidence suggests the major metabolic remodeling of the pathogen within different environmental conditions and its implications in therapy success [8]. Recent studies have demonstrated that, following phagocytosis, mycobacteria can control the metal concentrations inside the phagosome vacuole [3,17,18]. The concentrations of single metal elements have been established using X-ray microscopy in phagosomes of macrophages during pathogenic and non-pathogenic mycobacterial infections [17,18,19]. It was found that the iron concentration acquired by MAH and *M. tuberculosis* at 24 h of infection was significantly greater when compared to concentration levels at 1 h post-infection as well as to non-pathogenic and attenuated strains of mycobacteria [17], suggesting the ability of the pathogen to alter the intravacuolar concentration of specific metals for its own benefit. In fact, mycobacteria require iron for survival and growth within host cells and are known to produce siderophores and mycobactins to steal iron from the host iron-transporting proteins named transferrin [20,21,22]. 

In this study, we utilized a previously developed in vitro model mimicking the intracellular metal concentrations and pH encountered in MAH phagosomes [17] and investigated whether the intravacuolar environment of phagocytic cells could influence the development of the persistence state in mycobacteria and, thus, promoting the tolerance to currently available anti-MAH drugs.

## 2. Materials and Methods

### 2.1. Bacteria and Host Cells 

*Mycobacterium avium* strain 104 (MAH 104), a clinical isolate previously described [17], was grown in 7H9 Middlebrook liquid or on 7H10 Middlebrook agar supplemented with oleic acid-albumin-dextrose-catalase (OADC; Hardy Diagnostics, Santa Maria, CA, USA) for approximately 10 days at 37 °C. 

THP-1 monocytes (ATCC, TIB-202) were maintained in RPMI-1640 supplemented with 10% fetal bovine serum (FBS; Gemini Bio, Sacramento, CA, USA), L-glutamine, and at 37 °C in an atmosphere of 5% CO_2_. THP-1 cells were treated with 20 ng/mL of phorbol myristate acetate (PMA; Sigma-Aldrich, Saint Louis, MO, USA) overnight to differentiate them into macrophages. Monolayers of 5 × 10^5^ cells/well of 48-well tissue culture plate were infected with a multiplicity of infection (MOI) of 1. After 1 h of infection, cells were washed with hank’s balanced salt solution (HBSS; VWR, Visalia, CA, USA) three times and replenished with fresh RPMI-1640 medium. At selected time-points cells were lysed, and intracellular bacterial number were determined by counting the colony forming units (CFUs) on 7H10 Middlebrook agar plates and comparing to the original inoculum used for the infection. 

### 2.2. Antibiotic Susceptibility and Killing Kinetics In Vitro 

Antibiotics were purchased from Sigma-Aldrich (Saint Louis, MO, USA). MAH was exposed to a spectrum of concentrations (0, 1, 2, 4, 8, 16, 32, 64, 128, and 256 μg/mL) of clarithromycin (CLA), rifampicin (RIF), ciprofloxacin (CIP) and amikacin (AMK) to characterize MAH104 antibiotic susceptibility using a broth microdilution method. Briefly, 3 × 10^6^ CFU/mL bacteria were cultured in 7H9 Middlebrook broth with and without antibiotics and incubated in a shaker at 37 °C for one week. The minimal inhibitory concentration (MIC) and bactericidal concentration (BC) were visually determined at day 7. In addition, BC tubes were centrifuged at 10,000× *g*, resuspended in HBSS and plated on 7H10 agar plates to observe MAH104 viability, and the drug concentration that cleared 99.9% bacteria was considered as BC.

MAH104 inoculum of 3 × 10^6^ bacteria/mL was cultured in 7H9 liquid media supplemented with bactericidal concentrations of AMK (4 μg/mL), CLA (16 μg/mL), CIP (8 μg/mL) and RIF (32 μg/mL). Samples were tested for bacterial CFU counts every 24 h up to 7 days by plating the serial dilutions on 7H10 agar plates. 7H9 broth without antibiotics served as the bacterial growth control. 

### 2.3. MAH104 Growth in Presence of Metals and in the Dropout Metal Mix

All chemicals were purchased from Sigma-Aldrich (Saint Louis, MO, USA).The metal mix mimicking the intracellular metal concentrations and pH of MAH104 phagosomes at 1 h and 24 h post-infection, was prepared as previously described [17,23]. Briefly, metals listed in the Table 1 were added to 900 mL of Middlebrook 7H9 broth base and adjusted to pH 6.6 for the 1 h metal mix and pH 5.8 for the 24 h metal mix. Both mixtures were autoclaved and stored at 4 °C.

To establish MAH104 growth dynamics in Middlebrook 7H9 broth, the 1 h metal mix or the 24 h metal mix, approximately, 10^5^ bacteria of the mid-log phase grown MAH104 were inoculated into either 7H9 medium or metal mix of 1 h and 24 h and incubated at 37 °C for up to 4 days in a shaker. At each 24 h time-point, bacteria were diluted and plated for CFU quantification. Due to the fact that bacteria first will encounter 1 h phagosome metals and, later, 24 h, we also tested MAH104 growth of 1 h metal mix exposed bacteria in the 24 h metal mix. MAH104 was exposed to the 1 h metal mix for 1 h, centrifuged at 10,000× *g*, and bacterial pellet was resuspended and incubated in to the 24 h metal mix for additional 24 h. 

### 2.4. MAH104 Dose Response to Antibiotic Treatments before and after Exposure to Metal Mix

To address the question if metals of the phagosome environment can trigger persistence phenotype, the mid-log phase grown bacteria (10^5^ CFU/mL) were cultured into 5 mL of the Middlebrook 7H9 broth or the 24 h metal mix supplemented with or without antibiotics and incubated at 37 °C. The concentration range for all tested antibiotics (AMK, CLA, CIP and RIF) was 4–256 μg/mL. After 24 h, bacteria were centrifuged at 10,000× *g* for 20 min, pellets were further serially diluted and plated on the 7H10 agar plates supplemented with OADC. The plates were incubated at 37 °C for 10 days, and colonies were counted for CFU record of viable MAH104 recovered from the 7H9 broth or the metal mix under antibiotic pressure.

MAH104 persistence phenotype was also tested during the combinational treatment with 64 μg/mL CLA, 64 μg/mL CIP and 128 μg/mL RIF known to kill 99.9% MAH104 in 7H9 broth within 24 h. Approximately, 3 × 10^8^ bacteria from the mid-log phase were inoculated into either 7H9 medium, the 1 h metal mix or the 24 h metal mix with above combination of antibiotics at 37 °C for 24 h. We also tested MAH104 growth of 1 h metal mix exposed bacteria in the 24 h metal mix. The viable number of MAH104 without the combinational treatment in 7H9 broth, the 1 h, 1 h/24 h and 24 h metal mix served as controls. From each tested and control group, bacteria were centrifuged at 10,000× *g*, diluted and plated for CFU quantification.

In addition, in attempts to identify which metals may contribute to MAH104 persistence phenotype, approximately, 3 × 10^8^ bacteria from the mid-log phase were cultured into either 7H9 medium or dropout media (Table 1) in the 24 h metal mix base and subjected to the combinational treatment with 64 μg/mL CLA, 64 μg/mL CIP and 128 μg/mL RIF. No antibiotic treatment within tested condition served as a control.

### 2.5. Macrophage Infection and Response to TNF-α and IFN-γ Activation

MAH104 was exposed to the 1 h or the 24 h metal mix, each at corresponding time, while the 7H9 broth exposed bacteria were used as a control. MAH104 of these three phenotypes were used for inoculum preparation. THP-1 macrophages (5 × 10^5^ cells/well) in the 48-well tissue culture plate were infected at MOI of 10 for 1 h, washed twice with HBSS and replenished with fresh RPMI media containing 100 U/mL human TNF-α or 50 U/mL of IFN-γ (ThermoFisher Scientific, Waltham, MA, USA) [24]. The treatment was repeated at 24 h in fresh RPMI media. The monolayer viability was monitored daily to ensure that there was no toxic effect. Cells were lysed at 48 h post-infection with 0.1% Triton X-100 for 15 min, diluted and plated for quantification of bacterial CFU.

### 2.6. Statistical Analysis

Data analysis was performed using version 7 of GraphPad Prism, and statistical significance between the experimental and/or control groups was determined by the Student’s *t*-test. Experiments were repeated three times in duplicates unless otherwise indicated. *p* < 0.05 was considered significant.

## 3. Results

### 3.1. Susceptibility and Growth Dynamics of MAH104 to Antibiotics

To determine MAH 104 susceptibility to AMK, CLA, CIP and RIF, approximately, 1 × 10^5^ bacteria were exposed to antibiotic concentrations ranging 0–256 μg/mL using the broth microdilution method. The concentration at which 90% of MAH104 growth was inhibited by the antibiotic in the liquid culture at day 7 was considered as the minimum inhibitory concentration (MIC) and was recorded with the optical density (OD readings); whereas, the concentration at which 99.9% of MAH104 growth was inhibited by the antibiotic was considered as the bactericidal concentration (BC) and was recorded with the CFU assay on the 7H10 agar plates (Table 2).

Bacterial killing dynamics over 7 days were investigated for each antibiotic in the aerobic condition and under slight agitation in the shaking incubator at 37 °C (Figure 1). While bacterial exponential growth was seen over time in the 7H9 control group without antibiotic treatment, a complete bacterial clearance measured with MAH104 CFU counts was recorded at day 4 following AMK, CLA, CIP or RIF treatment.

### 3.2. MAH104 Growth Curve in 7H9 Broth, 1 h and 24 h Metal Mix

To determine if our in vitro model, mimicking the pH and metals encountered by MAH104 within the phagosome environment at 1 h and 24 h post-infection, has any influence on bacterial replication, 8 × 10^4^ MAH104 from the mid-log phase growth were cultured either in 7H9 broth, 1 h, 1 h/24 h- or 24 h metal mix. The bacterial growth dynamics was monitored over 4 days at 37 °C under slight agitation. Bacterial concentration, as measured by CFU/mL of culture, was recorded after each 24 h timepoint for 4 days and compared to 7H9 control. Results presented in the Figure 2 indicate that, in presence of 1 h phagosome metal mix, MAH104 growth was significantly delayed when compared to both 1 h/24 h metal mix and 24 h metal mix. While number of bacteria has increased in 1 h/24 h and 24 h metal mix in a similar manner, it was significantly slower than the 7H9 growth control, suggesting an inhibitory effect of phagosome metals on MAH104 (Figure 2).

### 3.3. Does MAH Exposure to an In Vitro Phagosome Model of Macrophage Lead to Selection of a Persistent Population?

To address the question, we investigated MAH104 response to AMK, CLA, CIP and RIF treatments at 24 h in wide-range concentrations of antibiotics before and after exposure to the 24 h metal mix (Figure 3). Results obtained for all antibiotic treatment groups demonstrate a greater increase in CFU for MAH104 exposed to the 24 h metal mix when compared to the corresponding drug concentration treatment in the 7H9 broth control. Data suggest a presence of the tolerant subpopulation of MAH104 at 24 h post antibiotic treatments in concentrations that significantly excided the drug bactericidal effect.

The incidence of persistent bacteria was also evaluated in the 1 h, 1 h/24 h and 24 h metal mix of the in vitro phagosome model during MAH104 combinational treatment with three antibiotics (CLA, CIP and RIF from different functional classes) (Figure 4). Results show that the prevalence of persistent bacteria counted in experimental groups after 24 h exposure with selected antibiotics was significantly higher than in the control (Figure 4A). In addition, to establish if treatment time had any influence in reducing the bacterial number, we cultured the 24 h metal mix exposed bacteria in presence of three antibiotics for up to 4 days (Figure 4B). Our results show that the MAH104 population become tolerant regardless of the exposure time and cannot be killed with high concentrations of CLA, CIP, and RIF employed. To eliminate the possibility of selection of the genetically resistant population of MAH 104, we isolated 30 single colonies, cultured in 7H9 broth and performed MIC testing for CLA, CIP and RIF. All selected colonies exhibit the similar MIC described for MAH104 (data not shown).

### 3.4. Exposure to In Vitro Phagosome Models of Metal Environment Promote MAH104 Survival within Activated Macrophages

To test if MAH104 exposure to 1 h and 24 h metal mix had an effect on intracellular bacterial growth within THP-1 cells, we incubated bacteria in both metal mix. The resulting population of MAH104 was then isolated and used for infection inoculum preparation. Our data indicate that MAH104 invasion rate at 1 h post-infection of THP-1 human macrophages was not influenced by different environments (Table 3). To determine whether exposure of MAH104 to 7H9 broth, 1 h and 24 h metal mix could affect bacterial killing by the host phagocytes upon activation by TNF-α or IFN-γ, we followed intracellular bacterial survival up to 48 h. These two cytokines are known to induce anti-MAH104 activity in macrophages [24]. The viable intracellular bacterial load was determined by plating lysed macrophages on 7H10 agar plates. The Table 3 shows that while the 1 h metal mix exposure slightly helped intracellular bacterial growth within THP-1 cells, the 24 h metal mix promoted a bacterial phenotype that significantly influenced MAH104 survival in activated macrophages by both cytokines versus the 7H9 broth control.

### 3.5. MAH104 Persistence in Metal Mix Lacking Specific Metals

To determine if a specific element had an effect on growth of MAH 104 in the 24 h mix, individual elements were removed from the complete media and bacterial CFU were recorded before and after combinational treatment with 64 μg/mL CLA, 64 μg/mL CIP and 128 μg/mL RIF. Bacterial growth in the complete 24 h metal mixt and the 7H9 broth were also recorded. The Figure 5 shows that while iron removal resulted in less persistent phenotype in the antibiotic treated MAH104 group, lack of manganese was associated with increased bacterial growth within the same antibiotic treated group. Observed changes were significant when compared to the bacterial number in the complete 24 h metal mix after antibiotic combination treatment. Results suggest that while iron enhances bacterial growth and weakens persistent state of bacteria, manganese, on the other hand, may enhance and support the development of persistent phenotype in MAH104.

## 4. Discussion

The main challenge of treating MAH patients is the inability of bactericidal antibiotics to eliminate infection quickly even with the use of combination therapy. When single antibiotics are used, development of drug resistance is inevitable [25]. The prolonged combination treatment results in favorable outcome in 40% to 60% of patients [26,27,28]. While some individuals might stay symptom-free for several years after therapy discontinuation, it has been shown that reinfection is common in this group of patients [29,30]. An incomplete response to therapy has been also associated with antimicrobials having declined efficacy against bacteria expressing the intracellular phenotype, while exhibiting complete potency against bacteria of in vitro culture [8]. During the intracellular cycle of infection, mycobacteria enter into low metabolic and nonreplicating state as an intrinsic response to cellular stress signaling and adverse environmental conditions such as nutrient starvation, low pH and hypoxia. These physiological changes contribute to bacterial intracellular fitness and persistence in the host and, at the same time, making the pathogen less susceptible to the action of antibiotics [8,31].

Persistence is a phenotypic trait in which bacteria exhibit a tolerance and, ultimately, protection to antibiotics, while remaining genetically susceptible [32]. A subpopulation of persistent bacteria has been isolated in numerous trials and has been characterized to have slow or stagnant growth [33,34,35]. The majority of antibiotics currently used for the treatment of MAH infections such as amikacin, clarithromycin, rifampicin, and ciprofloxacin, target actively growing cells [30,36]. Amikacin is an aminoglycoside antibiotic exhibiting bactericidal activity via binding to bacterial 30S ribosomal subunit and disrupting the protein synthesis process vital for growth. Clarithromycin is a bacteriostatic macrolide that binds to the 50S subunit of the bacterial ribosome and, therefore, acts by inhibiting the protein synthesis as well [30]. Rifampicin is also a bacteriostatic agent that belongs to the class of rifamycins and inhibits DNA dependent RNA polymerase with subsequent effect on protein synthesis [37]. Ciprofloxacin, a second-generation quinolone, binds to bacterial DNA gyrase and introduces the double-stranded breaks into DNA. Ciprofloxacin has been shown to have bactericidal activity against many pathogens including slow-growing bacteria [38], and in some cases selection of persistent bacteria has been reported [39]. The action of antibiotics against mycobacteria is more complex than initially thought. The recent study on MAH104 response to antibiotic treatments under conditions encountered on the surface of lung airways as well as intracellularly (such as anaerobic and biofilm) demonstrated the inability of AMK and CLA to significantly reduce MAH104 viability in vitro or clear infection within human macrophages [8]. Furthermore, the proteomic makeover of MAH104 under anaerobic and biofilm conditions identified some metabolic changes enhancing bacterial resistance to antibiotic treatment [8]. Recent study suggests a direct relationship between MAH104 metabolic state within biofilms and antimicrobial susceptibility, and improved antibiotic action against persistent forms of bacteria by enhancing its metabolism using metabolites [31].

Little is known about impact of phagosome elements on intracellular bacterial fitness and persistence mechanisms, despite that metals play an important role and are crucial for many cellular functions. Emerging literature provides strong evidence that pathogenic bacteria acquire metals from the host by competing with host nutritional defenses, and establishing a successful infection [21,40]. In an effort to distinguish if intracellular metals present within MAH104 phagosome of host macrophages can stimulate the persistence phenotype and the selection of tolerant bacteria against antibiotics, we utilized the in vitro model mimicking the metal concentrations and pH of MAH104 phagosome at 1 h and 24 h post-infection [17,23]. As expected within intracellular environment, both metal mixes exhibited an inhibitory effect on MAH104 replication in vitro. The significant delay in bacterial growth was observed in the 1 h phagosome metal mix than in the 24 h metal mix, however, the growth arrest of the pathogen was remarkable in both groups when compared to the 7H9 broth medium. Overall, while MAH104 number has increased in 24 h metal mix over time, we did not observe similar growth in the 1 h metal mix, suggesting more favorable adaptation to the environment at 24 h phagosome than at 1 h post-infection. The data also support the concept that during an adaptive response, MAH acquires metals at a concentration that support its viability and persistence within the host, while effluxing excess or toxic metals.

Furthermore, antibiotic treatment of metal mix exposed MAH104 demonstrates that metals present in the phagosome environment can enhance the persistence state, and greater number of tolerant bacteria is recovered from the 24 h metal mix when compared CFU numbers of the viable pathogen recovered from the 1 h metal mix and 7H9 broth. In addition, we observed a higher amount of tolerant subpopulation in all antimicrobial groups of the metal mix than in 7H9 broth, and these changes were noteworthy at the high concentration range as well as during combinational treatment of MAH104 over time. To establish if tolerance phenotype triggered by the metal mix can protect bacteria from killing mechanisms of macrophages, we infected human THP-1 cells with population of MAH104 recovered from the 1 h and 24 h metal mix, and tested the pathogen’s ability to survive in TNF-α and IFN-γ activated macrophages, shown to stimulate anti-MAH104 activity [24]. While we observed a slight increase in the intracellular growth of MAH104 of the 1 h metal mix, the survival rate in bacteria of the 24 h metal mix phenotype was significantly higher in activated macrophages by both cytokines when compared with the 1 h metal mix and 7H9 broth control, confirming presence of greater number of persistent MAH104 in the 24 h condition.

Mycobacteria can influence the intracellular environment through manipulation of the metal concentrations in the mycobacterial vacuole [18,19]. For instance, iron is seen in greater concentration in the mycobacterial vacuoles of cultured as well as mice macrophages at 24 h of infection than 1 h post-infection [17]. For establishing the intracellular niche and survival within the host, mycobacteria require iron [11,13,14]. The iron acquisition molecules such as siderophore and mycobactins assist bacteria to recruit transferrin, a host iron-transport protein, to the phagosome and, thus, increase the vacuolar iron concentration [41]. Our data show that in incomplete 24 h metal mix when iron is absent, MAH104 growth is significantly reduced by antibiotic treatment when compared to the complete 24 h metal mix alone. These results highlight a direct impact of iron on the ability of the pathogen to trigger the persistence mechanism, and further support a notion that the vacuolar iron is essential for mycobacterial pathogenicity in the host.

The mycobacterial vacuoles at 1 h and 24 h post-infection have detectable concentration of manganese [17,19]. The previous observation by our group demonstrates that manganese concentration at 1 h and 24 h post-infection of MAH vacuoles is seen at similar levels with no increase over time, and in both cultured and macrophages isolated from infected mice manganese concentration is significantly lower than the iron concentration [17,19]. Interestingly, when MnCl_2_ was removed from the 24 h metal mix, MAH104 growth was not affected as it seen for iron after the treatment with a combination of antibiotics. Moreover, manganese removal resulted in selection of MAH104 subpopulation with highest tolerance rate against antimicrobials when compared to all metal dropout groups during antibiotic treatment. Manganese pays a key role in adaptation of pathogenic bacteria in the host because it is the important micronutrient that pathogens utilize in the host to resist the effects of the oxidative stress [40,42]. Our data imply that MnCl_2_ presence in the metal mix constrains the formation of the persistent MAH104, which could be due to increased metabolic activity in bacteria in presence of manganese [40], therefore, making bacteria more susceptible to antibiotic treatment.

In summary, our study is the first report demonstrating the development of the persistence phenotype in mycobacteria under a pressure of phagosome metals and, subsequently, decreased effectiveness of antibiotic therapy. How individual metals promote MAH persistence and pathogenesis, in general, is not firmly established; however, our research for MAH and observations with *M. tuberculosis* [21,43] support a concept that the pathogenic mycobacteria regulate metal concentrations within the macrophage phagosome to promote an intracellular niche and persistence. Future research towards identification of components of metal acquisition machineries on the phagosome membrane and a better understanding of the pathogen mechanisms that control the metal sensing, assimilation or elimination from mycobacterial phagosomes will uncover new strategies to prevent a development of the persistence phenotype.

## Figures and Tables

**Figure 1 antibiotics-09-00927-f001:**
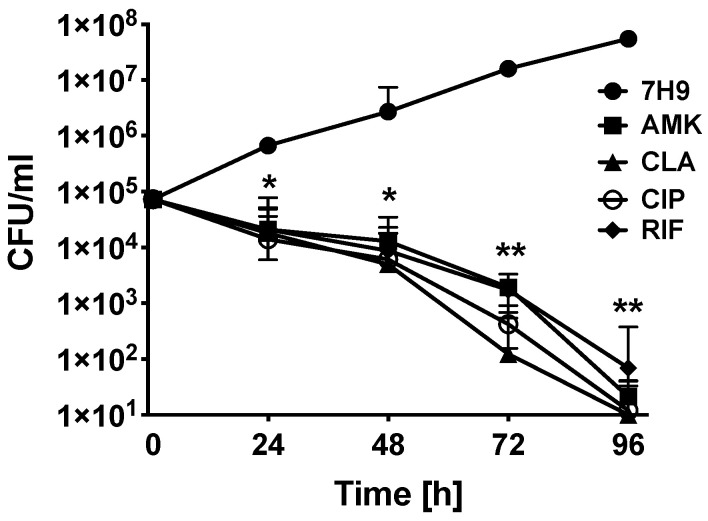
MAH104 in vitro killing kinetics by antibiotics. MAH104 time-kill curves using bactericidal concentrations of AMK, CLA, CIP, and RIF demonstrate viable bacterial colony forming units over 4 days. Antibiotics were added to bacterial cultures in 7H9 broth at time zero, and drug treatment groups were compared to MAH104 growth control. Data represent the means ± standard deviations (SD) obtained from three independent experiments. Unpaired two tailed *t*-test was performed. * *p* < 0.05 and ** *p* < 0.01 statistically significance between antibiotic treatment and control groups.

**Figure 2 antibiotics-09-00927-f002:**
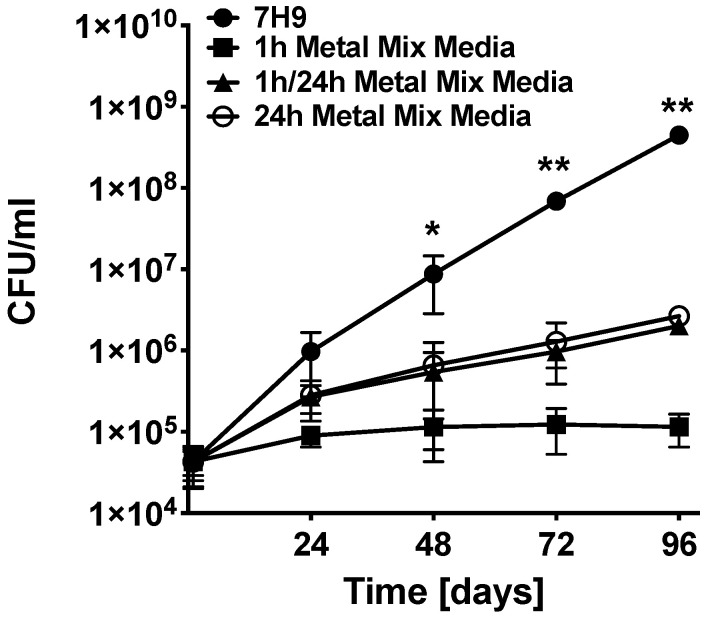
MAH104 growth dynamics in 7H9 broth culture and metal mix. The viable MAH104 colony forming units were recorded in 7H9 culture media or in vitro phagosome model of 1 h, 1 h/24 h or 24 h metal mix. Bacteria were cultured in the corresponding media at time zero and growth kinetics were monitored over 4 days. Data represent the means ± standard deviations (SD) obtained from four independent experiments performed in triplicates. Unpaired two tailed *t*-test was performed. * *p* < 0.05 and ** *p* < 0.01 statistical significance between the experimental (all metal mix) and control groups.

**Figure 3 antibiotics-09-00927-f003:**
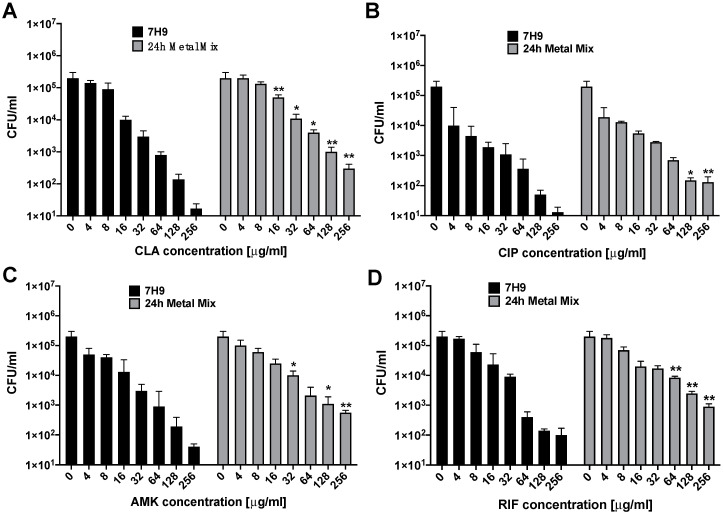
Antibiotic efficacy against MAH 104 in the 7H9 culture and the 24 h metal mix. The effect of the 24 h phagosome metal concentrations was evaluated by culturing MAH104 either in the 7H9 growth or the 24 h metal mix for 24 h. After, bacteria of two phenotypes were exposed to (**A**) CLA, (**B**) CIP, (**C**) AMK and (**D**) RIF in the concentration range of 4–256 μg/mL for additional 24 h at 37 °C. Next, bacteria were centrifuged, serially diluted and plated on 7H10 agar plates for CFU determination. Data represent the means ± standard deviations (SD) of three independent experiments performed in duplicates. * *p* < 0.05 or ** *p* < 0.01 statistical significance between the same drug concentration groups for bacteria exposed to the 7H9 broth and the 24 h metal mix.

**Figure 4 antibiotics-09-00927-f004:**
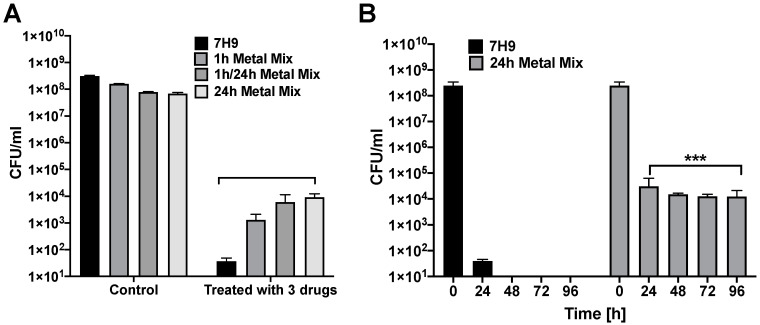
The persistent population of MAH104 produced during combinational antibiotic treatment. (**A**) Approximately, 5 × 10^8^ CFU/mL bacteria were cultured either in the 7H9 broth or the 1 h, 1 h/24 h and 24 h metal mix. After 24 h incubation, MAH104 of two phenotypes was subjected to 64 μg/mL CLA, 64 μg/mL CIP and 128 μg/mL RIF combinational treatment for additional 24 h at 37 °C under agitation. No antibiotic treatment within tested conditions served as controls. (**B**) MAH104 was incubated either in the 7H9 broth or the 24 h metal mix and subjected to 64 μg/mL CLA, 64 μg/mL CIP and 128 μg/mL RIF combinational treatment for up to 4 days. After, bacteria were centrifuged, serially diluted and plated on 7H10 agar plates for CFU determination. Data represent the means ± standard deviations (SD) of three independent experiments performed in duplicates. *** *p* < 0.001 statistical significance within the combinational antibiotic treatment between the 24 h metal mix and 7H9 control group.

**Figure 5 antibiotics-09-00927-f005:**
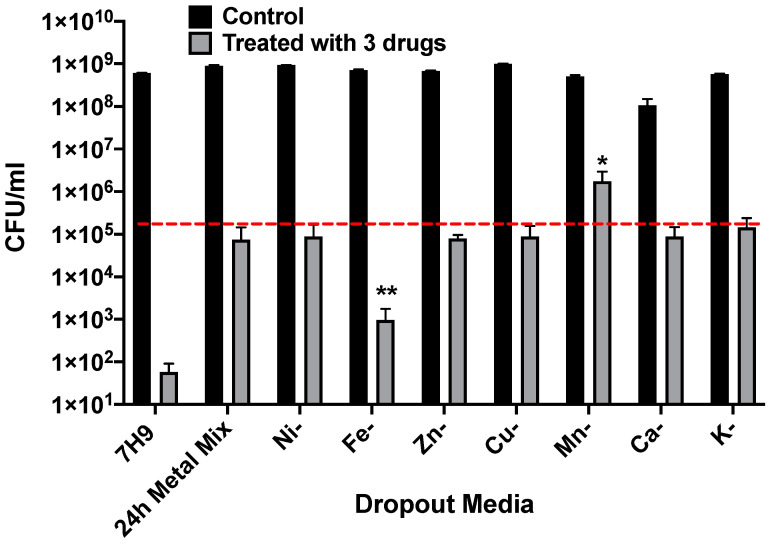
MAH104 growth in the 24 h metal mix missing elements. After initial 24 h incubation of bacteria in the incomplete 24 h metal mix missing single elements, bacteria were subjected to 64 μg/mL CLA, 64 μg/mL CIP and 128 μg/mL RIF combination treatment for additional 24 h at 37 °C under slight agitation. No antibiotic treatment within tested conditions served as a control. The dashed red line represents the viable MAH104 number recovered from the complete metal mix after combination treatment. Data represent the means ± standard deviations (SD) of three independent experiments performed in duplicates. * *p* < 0.05 and ** *p* < 0.01 statistical significance for the combinational antibiotic treatment between the incomplete and complete 24 h metal mix.

**Table 1 antibiotics-09-00927-t001:** Concentrations of single elements present in the metal mix and dropout media.

Supplements	Metal Mix (mL/L of 7H9 Broth)		Dropout 24 h Metal Mix (mL/L of 7H9 Broth)						
	1 h	24 h	-Ni	-Fe	-Zn	-Cu	-Mn	-Ca	-K
1 M potassium chloride (KCl)	14.7	0.925	0.925	0.925	0.925	0.925	0.925	0.925	-
1 M calcium chloride (CaCl_2_)	2	1.25	1.25	1.25	1.25	1.25	1.25	-	1.25
1 M manganese chloride (MnCl_2_)	5.9	11.9	11.9	11.9	11.9	11.9	-	11.9	11.9
1 M copper sulfate (CuSO_4_)	1.85	5.5	5.5	5.5	5.5	-	5.5	5.5	5.5
1 M zinc chloride (ZnCl_2_)	33	58.7	58.7	58.7	-	58.7	58.7	58.7	58.7
0.25 M ferric pyrophosphate (FePO_4_)	288	2	2	-	2	2	2	2	2
1 M nickel chloride (NiCl_2_)	5	5	-	5	5	5	5	5	5

**Table 2 antibiotics-09-00927-t002:** MAH104 susceptibility to antibiotics.

Antibiotic	*M. avium* subsp. *hominissuis* Strain 104	
	MIC (μg/mL)	BC (μg/mL)
Amikacin (AMK)	1	4
Clarithromycin (CLA)	1	16
Ciprofloxacin (CIP)	0.5	8
Rifampicin (RIF)	8	32

**Table 3 antibiotics-09-00927-t003:** MAH 104 survival in TNF-α and IFN-γ stimulated THP-1 cells.

Condition	Invasion (1 h)	Intracellular MAH104 (CFU/Well) at 48 h		
		No Treatment	TNF-α	IFN-γ
7H9	1.8 ± 0.3 × 10^5^	3.6 ± 0.4 × 10^5^	6.1 ± 0.3 × 10^3 (1)^	2.7 ± 0.3 × 10^4 (1)^
1 h Metal Mix	5.0 ± 0.6 × 10^5^	6.4 ± 06 × 10^5^	5.6 ± 0.4 × 10^4 (1,2)^	8.8 ± 0.3 × 10^4 (1,2)^
24 h Metal Mix	7.1 ± 0.4 × 10^4^	1.8 ± 0.4 × 10^5^	1.9 ± 0.3 × 10^5^	1.8 ± 0.5 × 10^5^

^(1)^*p* < 0.05 compared with no treatment control. ^(2)^
*p* < 0.05 compared with the 7H9 broth control.

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
