# Peer review of "Exposure of Mycobacterium avium subsp. homonissuis to Metal Concentrations of the Phagosome Environment Enhances the Selection of Persistent Subpopulation to Antibiotic Treatment"

_antibiotics, 2020, doi:10.3390/antibiotics9120927_

Round 1

Reviewer 1 Report

This is an interesting study on Mycobacterium avium subspecies hominissuis. It is well performed and the manuscript is well written. I have no scientific objection to the manuscript, just some grammatical amendments and a couple of slip of the pen mistakes.

Minor things:

Line 21: The treatment of MAH infections are prolonged ; change to: The treatment of MAH infections is prolonged

Line 34+35: greater number of tolerant bacteria are recovered; change to: greater number of tolerant bacteria is recovered

L 54: Mycobacteria is characterized; change to: Mycobacteria are characterized

L68: majority of the available antimicrobials target actively growing bacteria; change to: majority of the available antimicrobials targets actively growing bacteria

L 137-139: The metal mix…..were prepared; change to: The metal mix …..was prepared

Table 1: (Mn2Cl); change to: (MnCl2)

L 155: The concentration range….were; change to: The concentration range….was

L 183-184: statistical significance….were determined; change to: statistical significance….was determined

L 219-220: if our in vitro model…..have; change to: if our in vitro model….has

L 222: The number of bacterial; change to. The number of bacteria

L 223: Bacterial concentration……were recorded; change to: Bacterial concentration….was recorded

L 245: Does….leads to; change to: Does….lead to

L 269: Our results shows; change to: Our results show

L 270: hight; change to: high

L 278: MAH104 of two phenotypes were subjected; change to: MAH104 of two phenotypes was subjected

L 341: these group; change to: these groups

L 344: mycobacteria enters; change to: mycobacteria enter

L 362: selection of persistent bacteria have been reported; change to: selection of persistent bacteria has been reported

L 385: The data also supports; change to: The data also support

L 389+390: greater number of tolerant bacteria are recovered; change to: greater number of tolerant bacteria is recovered

L408: Our data shows; change to: Our data show

L 414+415: manganese concentration….are seen; change to: manganese concentration….is seen

L 421: Manganese pays a key role; change to: Manganese plays a key role

L 423: Our data implies; change to: Our data imply  

Author Response

The reviewer 1.

All changes are marked in red in the manuscript. Thank you for thorough reading of the paper and corrections.

This is an interesting study on Mycobacterium avium subspecies hominissuis. It is well performed and the manuscript is well written. I have no scientific objection to the manuscript, just some grammatical amendments and a couple of slip of the pen mistakes.

Minor things:

Line 21: The treatment of MAH infections are prolonged ; change to: The treatment of MAH infections is prolonged

            A: Now Line 18 “The treatment of MAH infections is prolonged and…”    

Line 34+35: greater number of tolerant bacteria are recovered; change to: greater number of tolerant bacteria is recovered

            A: Now Line 31-32 “greater number of tolerant bacteria is recovered from the 24h…”

L 54: Mycobacteria is characterized; change to: Mycobacteria are characterized

            A: Now Line 51” Mycobacteria are characterized with a thick…”

L68: majority of the available antimicrobials target actively growing bacteria; change to: majority of the available antimicrobials targets actively growing bacteria

            A: Line 65 “the available antimicrobials targets actively growing bacteria…”

L 137-139: The metal mix…..were prepared; change to: The metal mix …..was prepared

            A: Now Line 136 “infection, was prepared as previously described…”

Table 1: (Mn2Cl); change to: (MnCl2)

            A: This change is made in the Table 1.

L 155: The concentration range….were; change to: The concentration range….was

            A: Now Line 152 “The concentration range for all tested antibiotics (AMK, CLA, CIP and RIF) was…”

L 183-184: statistical significance….were determined; change to: statistical significance….was determined

            A: Now Line 181 “the experimental and/or control groups was determined by the Student’s t-test.”

L 219-220: if our in vitro model…..have; change to: if our in vitro model….has

            A: Lines: 217-218 “..1h and 24h post-infection, has any influence on bacterial replication..”

L 222: The number of bacterial; change to. The number of bacteria

            A: We changed the sentence. Now Line 219 “The bacterial growth dynamics was monitored…”

L 223: Bacterial concentration……were recorded; change to: Bacterial concentration….was recorded

            A: Now Line 220 “…by CFU/ml of culture, was recorded after…”

L 245: Does….leads to; change to: Does….lead to

A: Now Line 242 “Does MAH exposure to an in vitro phagosome model of macrophage lead to selection of a persistent population?”

L 269: Our results shows; change to: Our results show”

            A: Now Line 266 “Our results show that the MAH104…”

L 270: hight; change to: high

            A: Now Lines 267-268 “killed with high concentrations of…”

L 278: MAH104 of two phenotypes were subjected; change to: MAH104 of two phenotypes was subjected

            A: Now Line 275 “MAH104 of two phenotypes was subjected to…”

L 341: these group; change to: these groups

A: We made the following change on Line 338 “reinfection is common in this group of patients..”

L 344: mycobacteria enters; change to: mycobacteria enter

            A: Now Line 341 “..infection, mycobacteria enter into low..”

L 362: selection of persistent bacteria have been reported; change to: selection of persistent bacteria has been reported

            A: Now Line 359 “..persistent bacteria has been reported..”

L 385: The data also supports; change to: The data also support

            A: Now Line 382 “The data also support the concept..”

L 389+390: greater number of tolerant bacteria are recovered; change to: greater number of tolerant bacteria is recovered

            A: Now Line 387 “tolerant bacteria is recovered from the 24h metal mix..”

L408: Our data shows; change to: Our data show

            A: Now Line 405 “Our data show that in incomplete..”

L 414+415: manganese concentration….are seen; change to: manganese concentration….is seen

            A: Now Line 412 “..MAH vacuoles is seen at similar levels..”

L 421: Manganese pays a key role; change to: Manganese plays a key role

            A: Now Line 418 “Manganese pays a key role in adaptation..”

L 423: Our data implies; change to: Our data imply

            A: Now Line 420 “Our data imply that MnCl2 presence..”  

Reviewer 2 Report

The authors have used a previously developed in vitro model to identify metal concentrations in the phagosome that may influence the persistence phenotype of the MAH pathogen and render it less susceptible to antibiotic treatment. 

This an important work and could be further strengthened if the authors give more discussion on the implications of their research for those with M. avium infections,  rather than the "future research is needed..." line, even one sentence describing how these findings might be used to prevent the persistent phenotype would strength their concluding paragraph.

The manuscript is well-written, but has a few words that were omitted or misspelled. (Listed below).

  • Line 72 -- "5-10% of individuals.."
  • Lines 91-92 -- “...suggests the major metabolic remodeling.. occur? .. and has implications in therapy success”. This sentence is confused. Perhaps break into two sentences "for MAH (16). The experimental evidence suggests..."
  • Line 103 -- “utilized a previously developed 
  • Line 222 -- "The number of bacterial..." what? bacterial concentration was monitored?
  • Line 270-71 -- “hight contrations” misspelled
  • Line 373 -- "despite that metals play an important role..."
  • Line 385 -- "The data also support" 

Author Response

The reviewer 2.

All changes are marked in red in the manuscript. Thank you for a suggestion and corrections.

This an important work and could be further strengthened if the authors give more discussion on the implications of their research for those with M. avium infections,  rather than the "future research is needed..." line, even one sentence describing how these findings might be used to prevent the persistent phenotype would strength their concluding paragraph.

A: As suggested by the reviewer we modified summary paragraph: Lines 424-433

“In summary, our study is the first report demonstrating the development of the persistence phenotype in mycobacteria under a pressure of phagosome metals and, subsequently, decreased effectiveness of antibiotic therapy. How individual metals promote MAH persistence and pathogenesis, in general, is not firmly established; however, our research for MAH and observations with M. tuberculosis [21,43] support a concept that the pathogenic mycobacteria regulate metal concentrations within the macrophage phagosome to promote an intracellular niche and persistence. A future research towards identification of components of metal acquisition machineries on the phagosome membrane and a better understanding of the pathogen mechanisms that control the metal sensing, assimilation or elimination from mycobacterial phagosomes will uncover new strategies to prevent a development of the persistence phenotype.”

The manuscript is well-written, but has a few words that were omitted or misspelled. (Listed below).

  • Line 72 -- "5-10% of individuals.."

A: We changed it. Line 69. “It is a fact that, after infection, 5-10% of individuals develop tuberculosis disease..”

  • Lines 91-92 -- “...suggests the major metabolic remodeling.. occur? .. and has implications in therapy success”. This sentence is confused. Perhaps break into two sentences "for MAH (16). The experimental evidence suggests..."

A: We changed it. Lines 87. “The persistence phenomenon has been less investigated for MAH [16]. The experimental evidence suggests the major metabolic remodeling..”

  • Line 103 -- “utilized a previously developed 

A: We made this change. Line 100. “In this study, we utilized a previously developed in vitro model..”

  • Line 222 -- "The number of bacterial..." what? bacterial concentration was monitored?

A: We changed the sentence. Line 219. “The bacterial growth dynamics was monitored over 4 days at 37oC under…”

  • Line 270-71 -- “hight contrations” misspelled

A: We corrected it. Line 267-268. “killed with high concentrations of..”

  • Line 373 -- "despite that metals play an important role..."

A: We made a correction. Line 370. “despite that metals play an important role and are crucial for many cellular…”

  • Line 385 -- "The data also support" 

A: We corrected it. Line 382. “The data also support the concept that…”